# Evaluation of Rumen Fermentation and Microbial Adaptation to Three Red Seaweeds Using the Rumen Simulation Technique

**DOI:** 10.3390/ani13101643

**Published:** 2023-05-15

**Authors:** Stephanie A. Terry, Ana M. Krüger, Paulo M. T. Lima, Robert J. Gruninger, D. Wade Abbott, Karen A. Beauchemin

**Affiliations:** 1Lethbridge Research and Development Centre, Agriculture and Agri-Food Canada, Lethbridge, AB T1J 3B6, Canada; 2Centro de Energia Nuclear na Agricultura, Universidade de São Paulo, São Paulo 13400-970, Brazil

**Keywords:** rumen simulation technique, methane production, seaweed, rumen fermentation

## Abstract

**Simple Summary:**

There is a global focus on the search for identifying species of macroalgae that can be fed to livestock as a feedstock, or as a methane mitigant. This study evaluated the effect of three red seaweeds to assess their potential for decreasing ruminant enteric methane production. While only *Asparagopsis taxiformis* was effective at decreasing methane production, this study provides new information on the requirements for adapting animals to seaweed feeding.

**Abstract:**

Several red seaweeds have been shown to inhibit enteric CH4 production; however, the adaptation of fermentation parameters to their presence is not well understood. The objective of this study was to examine the effect of three red seaweeds (*Asparargopsis taxiformis*, *Mazzaella japonica*, and *Palmaria mollis*) on in vitro fermentation, CH4 production, and adaptation using the rumen simulation technique (RUSITEC). The experiment was conducted as a completely randomized design with four treatments, duplicated in two identical RUSITEC apparatus equipped with eight fermenter vessels each. The four treatments included the control and the three red seaweeds added to the control diet at 2% diet DM. The experimental period was divided into four phases including a baseline phase (d 0–7; no seaweed included), an adaptation phase (d 8–11; seaweed included in treatment vessels), an intermediate phase (d 12–16), and a stable phase (d 17–21). The degradability of organic matter (*p* = 0.04) and neutral detergent fibre (*p* = 0.05) was decreased by *A. taxiformis* during the adaptation phase, but returned to control levels in the stable phase. *A. taxiformis* supplementation resulted in a decrease (*p* < 0.001) in the molar proportions of acetate, propionate, and total volatile fatty acid (VFA) production, with an increase in the molar proportions of butyrate, caproate, and valerate; the other seaweeds had no effect (*p* > 0.05) on the molar proportions or production of individual VFA. *A. taxiformis* was the only seaweed to suppress CH4 production (*p* < 0.001), with the suppressive effect increasing (*p* < 0.001) across phases. Similarly, *A. taxiformis* increased (*p* < 0.001) the production of hydrogen (H2, %, mL/d) across the adaptation, intermediate, and stable phases, with the intermediate and stable phases having greater H2 production than the adaptation phase. In conclusion, *M. japonica* and *P. mollis* did not impact rumen fermentation or inhibit CH4 production within the RUSITEC. In contrast, we conclude that *A. taxiformis* is an effective CH4 inhibitor and its introduction to the ruminal environment requires a period of adaptation; however, the large magnitude of CH4 suppression by *A. taxiformis* inhibits VFA synthesis, which may restrict the production performance in vivo.

## 1. Introduction

Increased greenhouse gas (GHG) concentrations in the atmosphere have resulted in alterations to the ozone layer, consequently raising the global surface temperature [1]. The agricultural sector contributes to 26.0% of anthropogenic global GHG, mainly CO2, N2O, and CH4 [2]. Enteric CH4 is produced from the natural fermentation of carbohydrates within the rumen, and contributes to approximately 6.0% of the global anthropogenic GHG [3]. While CH4 has a comparatively shorter half-life in the atmosphere (~10 years) than CO2 which can persist in the atmosphere for hundreds of years, CH4 has 28 times the global warming potential of CO2, making it an attractive target for abatement [4].

There has been a growing interest in the use of macroalgae or seaweed and their associated by-products to reduce enteric CH_4_ emissions from ruminants [5]. Macroalgae are rich in complex carbohydrates and polysaccharides, including two groups of compounds that are known CH_4_ inhibitors. Specific to red seaweeds are the presence of halogenated low molecular weight compounds, including bromoforms and haloforms [6,7]. These compounds have been shown to be extremely effective at mitigating enteric CH_4_ production with reports of greater than 67% reduction observed with the feeding of the *Asparagopsis* species to various ruminants [6,7]. Secondly, seaweeds contain a variety of phlorotannins that are similar to their terrestrial counterparts [5]. Terrestrial tannins have been shown to decrease enteric CH_4_ production by as much as 30%, with the extent varying dramatically based on tannin profiles [8]. It is hypothesised that seaweeds with high concentrations of phlorotannins may exhibit a similar antimethanogenic property to terrestrial tannins.

Rumen adaptation to feed additives such as tannins, fats, and chemical inhibitors has been well studied; however, knowledge about the ruminal adaptation dynamics to seaweed supplementation is still required. Therefore, the objective of this study was to examine the effect of the three red seaweeds *Asparagopsis taxiformis*, *Mazzaella japonica*, and *Palmaria mollis* on in vitro fermentation and gas production, and evaluate the adaptation response of fermentation characteristics to seaweed supplementation within a rumen simulation technique (RUSITEC) system fed a barley straw and silage diet.

## 2. Materials and Methods

The experiment was conducted at Agriculture and Agri-Food Canada in Lethbridge, AB, Canada. Donor heifers used in this experiment were cared for in accordance with the guidelines of the Canadian Council on Animal Care (2009).

### 2.1. Seaweed

The three seaweeds used in this experiment included *A. taxiformis*, *M. japonica*, and *P. mollis*. The seaweeds were chosen based on their biomass availability, harvest potential, and biochemical composition. *A. taxiformis* was chosen based on its previously observed characterisation to manipulate rumen fermentation and CH_4_ production and was assigned as a positive control. The novel seaweeds *M. japonica* and *P. mollis* are wild harvested off the coast of British Columbia, Canada; *M. japonica* mainly for its carrageenan content and *P. mollis* for human consumption.

### 2.2. Experimental Design and Treatments

The experiment was conducted as a completely randomized design with four treatments, duplicated in two identical RUSITEC apparatus equipped with eight fermenter vessels each. The four treatments included the control (no added seaweed), and three red seaweeds (*A. taxiformis*, *M. japonica*, *P. mollis*) included at 2% diet DM. The substrate consisted of a 50:50 barley straw and barley silage diet (DM basis). The chemical compositions of the substrates and seaweeds are shown in Table 1. An elemental analysis of the seaweeds are shown in Table 2.

The experimental period was divided into four phases: a baseline phase (d 0–7) where fermenters were only fed the diet substrate and measurements were only recorded on d 5–7; the adaptation phase in which seaweed was introduced into the allocated diets and measurements were recorded from d 8–11; the intermediate phase (d 12–16); and the stable phase (d 17–21) to assess the changes in rumen fermentation and the microbial populations in response to feeding on seaweed (Figure 1).

### 2.3. Substrate Processing

Barley straw, barley silage, and seaweeds were ground through a 4 mm screen using a Wiley mill (standard model 4; Arthur H. Thomas Co., Philadelphia, PA, USA). A total of 10 g diet DM was fed to each fermenter daily in bags (10 × 20 cm; 50 ± 10 µ porosity; R1020, ANKOM Technology, Macedon, NY, USA). For the control and during the baseline phase, 5 g DM each of barley straw and barley silage were included in the bags. For the seaweed treatments, 0.2 g DM of seaweed replaced equal proportions of barley straw and barley silage.

### 2.4. Inoculum Sampling and Incubation Procedure

Two RUSITEC apparatuses each fitted with eight fermenters (920 mL) were used for the in vitro incubation, so that each treatment was randomly allocated to two fermenters within each apparatus (*n* = 4 vessels/treatment). Each fermenter was fitted with a site for artificial saliva infusion and effluent output. Rumen inoculum was obtained from three ruminally cannulated beef heifers previously adapted for two weeks to a barley straw and barley silage diet which included a mineral supplement. Rumen fluid and solid contents were pooled from the three heifers, filtered through four layers of cheesecloth, and transported to the laboratory in an insulated thermos. The fermenters were maintained at 39 °C by immersion in a water bath. Each fermenter was pre-filled with 180 mL of pre-warmed McDougall’s buffer (pH = 8.2; [9])and 720 mL of strained rumen fluid.

One R1020 bag (ANKOM Technology) containing 20 g of mixed solid rumen digesta, and one bag containing 10 g DM of the diet were allocated to each fermenter. After 24 h, the bag containing rumen digesta was replaced by a bag containing the diet. Thereafter, one bag was replaced daily so that each bag remained in the fermenter for 48 h. Bags containing the seaweed treatments were introduced into the fermenters on d 8.

The artificial saliva was continually infused into the fermenters using a peristaltic pump set to achieve a dilution rate of 2.9%/h. The effluent was collected in a 1 L flask, and gas was collected in a 2 L gas tight bag (Curity^®^; Conviden Ltd., Mansfield, MA, USA) attached to the effluent flask. Feed bag exchange, fermenter pH, gas production, and effluent volume were measured every 24 h at 10 am.

### 2.5. Nutrient Disappearance

Dry matter degradability (DMD) was determined from d 3 to 21 after 48 h of fermentation. The feed bags were removed, washed in cold running water for 2 min, and dried at 55 °C for 48 h ([10]; method 930.15) for the determination of DMD. After drying, the residues were pooled from d 9 to 11, and d 17 to 19, ground through a 1 mm screen (Wiley mill, standard model 4; Arthur H. Thomas Co., Philadelphia, PA, USA), and analysed for organic matter (OM), NDF, CP, and ether extract (EE) concentrations. This generated samples representing the adaptation and stable phase. Insufficient samples were available for the baseline and intermediate phases, as the samples were used for a microbial profiling analysis, published elsewhere [11].

The samples were dried at 550 °C for 5 h and OM was calculated as 100—ash ([10]; method 942.05). The NDF content was determined using an ANKOM200 Fibre Analyser based on the procedure described by Van Soest, et al. [12] using sodium sulphite and α-amylase as reagents and expressed exclusive of residual ash. The total N concentration was quantified by flash combustion with gas chromatography and thermal conductivity detection (Carlo Erba Instruments, Milan, Italy [13]; method 990.03). The CP content was calculated as the N concentration × 6.25. Fat was determined according to AOAC (2006; method 2003.05) using ether extraction (Extraction Unit E-816 HE; Büchi Labortechnik AG, Flawil, Switzerland).

### 2.6. Gas Production

The total gas production was determined daily using a gas meter (Model DM3A, Alexander-Wright, London, England, UK). A 20 mL sample was collected from the septum of the collection bag using a 26-gauge needle and transferred to a pre-evacuated exetainer (6.8 mL; Labco Ltd., Wycombe, Buckinghamshire, UK). Concentrations of CH_4_, O_2_, H_2_, and CO_2_ were determined using a gas chromatograph equipped with a GS-Carbon-PLOT (30 m × 0.32 mm × 3 mm) column and thermal conductivity detector (Agilent Technologies Canada, Inc., Mississauga, ON, Canada) at an isothermal oven temperature of 35 °C, with He as the carrier gas (27 cm/s).

### 2.7. Fermentation Variables

Effluent volume and fermenter pH was recorded daily at the time of feed bag exchange. Two 5 mL effluent samples were placed in vials prefilled with 1 mL of 25% (*wt*/*vol*) metaphosphoric acid and 1 mL of 1% (*wt*/*vol*) sulphuric acid for an analysis of volatile fatty acid (VFA) composition and NH_3_ concentration, respectively. Sample vials were kept at −20 °C until analysis.

The concentration of VFA was determined by gas chromatography (5890A Series Plus II, Hewlett Packard Co., Palo Alto, CA, USA) equipped with a 30 m Zebron free fatty acid phase fused silica capillary (0.32 mm i.d., 1.0 μm film thickness; Phenomenex, Torrance, CA, USA). The concentration of NH_3_ was determined using the phenol-hypochlorite method as described by Broderick and Kang [14].

### 2.8. Elemental Analysis

The elemental analysis of seaweeds was conducted by a commercial laboratory (Cumberland Valley Analytical Services, Waynesboro, PA, USA). Phosphorus, Ca, Mg, K, Na, Fe, Mn, Zn, and Cu were determined using the AOAC method 985.1 [15] with modifications where seaweeds were ashed for 1 h at 535 °C, digested in open crucibles for 20 min in 15% HNO_3_ on a hotplate, diluted to 50 mL, and then analysed using inductively coupled plasma. The samples used for the Mo analysis were ashed at 480 °C for 4 h, digested in an open crucible for 20 min in 15% HNO_3_ on a hotplate, diluted to 50 mL, and then analysed on axial view inductively coupled plasma. The selenium was analysed using the AOAC method 996.16 [15]. The bromoform concentration of the seaweed was conducted by a commercial laboratory (Bigelow Analytical Services, East Boothbay, ME, USA) using methanol extraction with samples analysed by GC/MS.

### 2.9. Calculations and Statistical Analysis

The disappearance of DM, OM, NDF, and CP (DMD, OMD, NDFD, and CPD) was calculated as the difference between the nutrient content before and after incubation, and expressed as a percentage. Methane production was expressed as mg per g of DM digested (DMd) and incubated (DMi).
DMd = Total Dmi − Total DM remaining after incubation

Data were analysed using the MIXED model procedure of SAS (SAS Inc., Cary, NC, USA). Individual fermenter was considered the experimental unit with the day of sampling treated as a repeated measure. Treatment, phase, and treatment × phase were considered as fixed effects while fermenter within the vessel was considered as a random effect. Minimum values of Akaike’s information criterion were used to select the covariance structure. Data were tested for normality of variance. Significance was declared at *p* ≤ 0.05.

## 3. Results

There was a treatment × phase effect (*p* ≤ 0.05) on OMD, NDFD, propionate, branched-chain VFA (BCVFA), caproate, valerate, acetate:propionate molar proportions, and total VFA production (Table 3). When averaged across all phases, DMD was greatest (*p* < 0.01) for *A. taxiformis* and lowest for *P. mollis*; however, neither were different from the control (*p* > 0.05). There was no effect (*p* > 0.05) of treatment on CPD, pH, or NH_3_ production. Effluent output was greater (*p* = 0.03) in *P. mollis* than *A. taxiformis*, although neither were different from the control (*p* > 0.05). Both OMD (*p* = 0.04) and NDFD (*p* = 0.05) were decreased by *A. taxiformis* during the adaptation phase, but returned to control levels in the stable phase (Figure 2).

The molar proportions of acetate were decreased (*p* < 0.001) and butyrate increased (*p* < 0.001) by *A. taxiformis* compared with all other treatments. *A. taxiformis* decreased the molar proportions of propionate (*p* < 0.001) during the adaptation, intermediate, and stable phases with the decreases being largest in the intermediate and stable phases (Figure 3a). There was no effect of *M. japonica* or *P. mollis* on total VFA across any phase; however, *A. taxiformis* had less (*p* = 0.01) total VFA during the intermediate and stable phases compared with the other treatments. The acetate:propionate was higher in *A. taxiformis* than the *M. japonica* and *P. mollis* treatments during adaptation, although it was not different from the control during this phase (Figure 3b). *A. taxiformis* increased (*p* < 0.001) molar proportions of caproate, valerate, and BCVFA during the intermediate and stable phase, with no difference detected during the adaptation phase (Figure 3b).

There was no effect (*p* ≥ 0.34) of treatment on gas or O_2_ production (% or mL; Table 4). *A. taxiformis* decreased (*p* < 0.05) CO_2_ (%, mL/d) production compared to *P. mollis*, but neither was different from the control (*p* > 0.05). There was a treatment × phase effect (*p* < 0.001) for CH_4_ (%, mL/d, mg/d, mg/g DMd, mg/g DMi) and H_2_ (%, mL/d). The control, M. japonica, and *P. mollis* had similar CH_4_ production across all phases (Figure 4a,b), whereas *A. taxiformis* decreased (*p* < 0.001) CH_4_ production across the adaptation, intermediate, and stable phases compared with other treatments. The production of H_2_ (%, mL/d) was increased (*p* < 0.001) across the adaptation, intermediate, and stable phases by *A. taxiformis*, with the intermediate and stable phases having greater H_2_ production than the adaptation phase (Figure 5).

## 4. Discussion

This study examined the effect of three red seaweeds on fermentation, nutrient degradability, and gas production within a RUSITEC fed a roughage-based diet. The novelty in this study was the examination of the adaptation of the fermentation system to the introduction of three different seaweeds (*A. taxiformis*, *M. japonica*, and *P. palmate*). A 7 d period was allocated for system adaptation (baseline) as well as examining the fermentation parameters before the introduction of seaweeds. Thereafter, the measurements were divided into the adaptation (d 8–11), intermediate (d 12–16), and stable phases (d 17–21) to evaluate the changes that occurred after administrating the seaweeds until the system and the microbial population became more stabilised.

The lack of difference in DMD between the control and each seaweed demonstrates the ability of the microbial population to effectively degrade seaweed carbohydrates, with all treatments improving in DMD across the different phases indicating that the microbial population became more efficient over time [11]. Compared with DMD, OMD and NDFD were more sensitive to the addition of *A. taxiformis* when introduced to the fermenters, given the initial decrease in both variables during the adaptation phase. The disruption to the fermentation of feed due to the introduction of *A. taxiformis* may have been caused by the radical suppression of CH_4_, the increase in H_2_, and an overall shift in the metabolome requiring the adaptation of the microbiome. Conversely, the recovery of OMD and NDFD in the stable phase indicated that only a relatively short period of time was required for the rumen microbial community to adapt to the presence of *A. taxiformis*. This short disruption in feed degradation was unique to *A. taxiformis* and was not observed with either *M. japonica* or *P. mollis*.

The suppression of CH_4_ production by *A. taxiformis* has been previously documented by both in vitro and in vivo studies [5,7,16,17,18,19]. The present study also observed a rapid drop in CH_4_ production with *A. taxiformis* inclusion throughout the adaptation, intermediate, and stable phases compared to the control. The drop in CH_4_ production (mL/g DMd) compared to the control was 80.2, 93.7, and 95.1% over the adaptation, intermediate and stable phases, respectively. Although not significantly different between these phases, the results suggest that there was an adaptation period to the introduction of *A. taxiformis*, with the CH_4_ suppressing effect increasing over time with continued *A. taxiformis* addition. However, within a RUSITEC system, it is frequent to observe reductions in certain populations of the microbiota as the length of the experiment increases. For example, Mateos, et al. [20] found that in solid associated samples, protozoal DNA concentration and an abundance of *Fibrobacter succinogenes*, *Ruminococcus albus*, and fungi decreased, and the abundance of methanogenic archaea increased over a 14 d period within a RUSITEC, despite relatively stable fermentation variables [20]. The changes in microbiota over time may also explain the differences observed within the same treatment over time, because although there were significant decreases in CH_4_ metrics observed throughout the phases for *M. japonica* and *P. mollis*, within each phase they were not different from the control [11]. The lack of effect of these two species on CH_4_ production is reinforced by consistent H_2_ production observed across all phases. Similarly, in a companion study in our laboratory using a batch culture technique (unpublished data), we observed that neither *M. japonica* nor *P. mollis* had an effect on in vitro fermentation or CH_4_ production in a barley straw diet. The ineffectiveness of these two seaweeds on reducing CH_4_ production is likely due to the lack of concentrations of bromoforms in these species. Bromoform has been verified as the effective component in some seaweeds that inhibit enteric CH_4_ production [7].

The impact of including *A. taxiformis* within a RUSITEC has been previously evaluated; however, measurements were only taken from the immediate addition of the seaweed and at 4, 12, and 24 h intervals each day over a period of 4 days in that study [18]. Roque, Brooke, Ladau, Polley, Marsh, Najafi, Pandey, Singh, Kinley, Salwen, Eloe-Fadrosh, Kebreab and Hess [18] observed that the 5% OM inclusion rate of *A. taxiformis* decreased CH_4_ production by 95%, with the suppression of CH_4_ production observed at the first sampling that was conducted 28 h after seaweed introduction into the system. Methane production in that study was almost zero after 76 h of incubation, although no further measurements were observed past this time. In contrast, our experiment showed that the ability of *A. taxiformis* to decrease CH_4_ improved over time with some CH_4_ production still observed (~1.35 mg/d) during the final phase. The decrease in CH_4_ production from *A. taxiformis* in the present study is consistent with an increase in H_2_ production over the phases, with the greatest CH_4_ suppression observed in the intermediate and stable phases resulting in more H_2_ production than in the adaptation phase. This is further verified by a companion study that found that the inclusion of *A. taxiformis* had a significant impact on the microbiome, including a large reduction in all major archaeal species [11].

Although the total gas production was not affected by seaweed treatment, *A. taxiformis* caused a reduction in total VFA production in the adaptation, intermediate, and stable phases, demonstrating a reduction in the microbial degradation of nutrients. Organic matter is degraded by the rumen consortium, generating VFA, the main source of energy provided to ruminants. Iso-acids are products from the degradation of valine, isoleucine, leucine, and proline which are used in the biosynthesis of higher BCVFA [21]. The BCVFA are required for optimal fibre degradation and efficiency of ruminal fermentation [22]. The increase in these intermediates (butyrate, caproate, BCVFA, and valerate) corresponds with the decreased production of acetate and propionate indicating that *A. taxiformis* inhibited major VFA synthesis. This conclusion is verified by O’Hara [11], who demonstrated that *A. taxiformis* inhibited major VFA producing bacteria including species of *Fibrobacter* and *Ruminococcus*. Other chemical CH_4_ inhibitors that have resulted in large decreases (7–29%) of CH_4_ have also been found to result in the increased production of valerate and isovalerate [23,24], consistent with the hypothesis that increased rumen H_2_ favours the fermentation pathways that consume H_2_, including valerate and caproate [25,26].

Inhibiting CH_4_ production can theoretically increase the availability of H_2_ for incorporation into VFA, thereby increasing energy availability to the animal [26,27]. Yet, despite the increase in H_2_ production that accompanied the decrease in CH_4_ production for the *A. taxiformis* treatment in the present study, the total VFA was not increased. Furthermore, an in vitro study found that at 1, 2, and 5% OM inclusion of *A. taxiformis*, total VFA was decreased by 16.6, 25.0, and 39.5%, respectively [17]. Machado, Magnusson, Paul, Kinley, de Nys and Tomkins [17] also observed alterations in the molar proportions of VFA with propionate, butyrate, valerate, and isovalerate increasing and acetate and isobutyrate decreasing compared with the control. In contrast, Roque, Brooke, Ladau, Polley, Marsh, Najafi, Pandey, Singh, Kinley, Salwen, Eloe-Fadrosh, Kebreab and Hess [18] did not observe a significant decrease in total VFA with *A. taxiformis*, but did find a decrease in the acetate:propionate ratio and valerate production, in comparison with the current study where valerate production increased with the inclusion of *A. taxiformis*. Valerate along with caproate are intermediary VFA and the increase with *A. taxiformis* is related to its lack of incorporation into the three main VFA, indicating an inefficiency of fermentation possibly brought by the decrease in H_2_ incorporation into CH_4_ [21]. The increase in BCVFA, valerate, and caproate may also indicate that ruminal microbial growth is not optimised in the presence of *A. taxiformis*, as these VFA are essential for cellulolytic bacteria growth, which may contribute towards the reduced NDFD during adaptation.

## 5. Conclusions

In conclusion, this study found that *M. japonica* and *P. palamata* did not impact rumen fermentation or exhibit a CH_4_-suppressing capacity. In contrast, *A. taxiformis* was shown to be an effective CH_4_ suppressant with its immediate addition causing negative alterations to fermentation variables, with the large magnitude of CH_4_ suppression inhibiting VFA synthesis. These findings may indicate that feeding *A. taxiformis* to ruminants at a dose rate that results in a large decrease in CH4 production may alter rumen metabolism in a manner that restricts production performance through reduced VFA synthesis.

## Figures and Tables

**Figure 1 animals-13-01643-f001:**
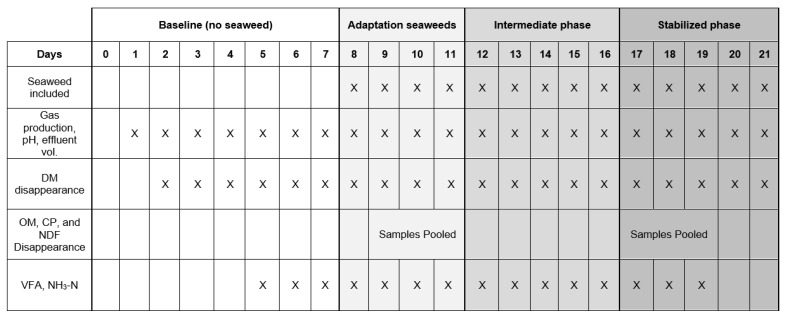
Outline of experiment dosing and sampling.

**Figure 2 animals-13-01643-f002:**
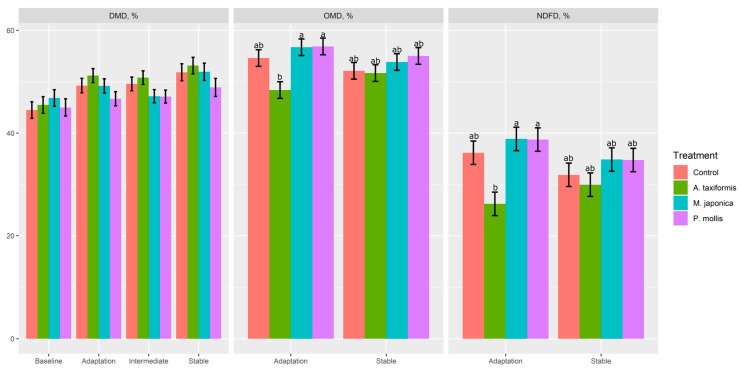
The effect of seaweed and phase on dry matter disappearance (DMD), organic matter disappearance (OMD), and neutral detergent fibre disappearance (NDFD) in a RUSITEC (*n* = 4). ^a,b^ Within variable, means without a common superscript differ (*p* ≤ 0.05); variables without letters do not have a significant (*p* > 0.05) treatment × phase effect.

**Figure 3 animals-13-01643-f003:**
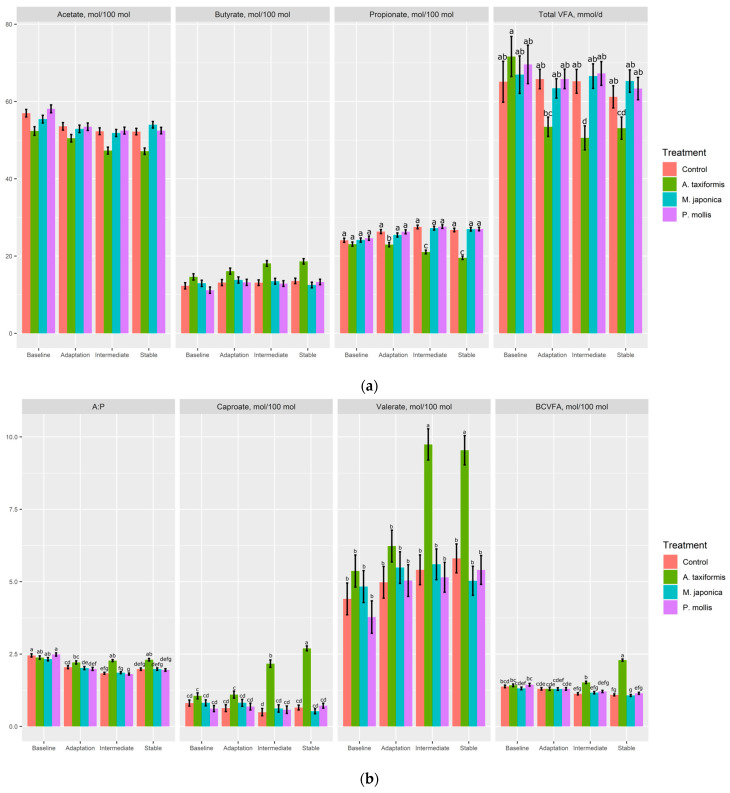
The effect of seaweed and phase on the (**a**) molar portions of acetate, propionate, butyrate, and total volatile fatty acids; and (**b**) acetate:propionate ratio, and molar proportions of caproate, branched-chain volatile fatty acids (BCVFA), and valerate (*n* = 4). ^a–g^ Within variable, means without a common superscript differ (*p* ≤ 0.05); variables without letters do not have a significant (*p* > 0.05) treatment × phase effect.

**Figure 4 animals-13-01643-f004:**
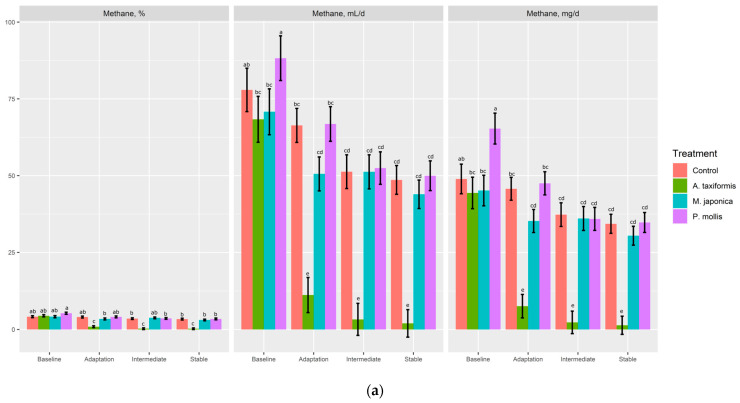
The effect of seaweed and phase on the (**a**) percentage of CH_4_, and CH_4_ production (mL/d, mg/d), and (**b**) CH_4_ production on a dry matter disappearance (mg/g DMd) and dry matter incubated basis (mg/g DMi) (*n* = 4). ^a–d^ Within variable, means without a common superscript differ (*p* ≤ 0.05).

**Figure 5 animals-13-01643-f005:**
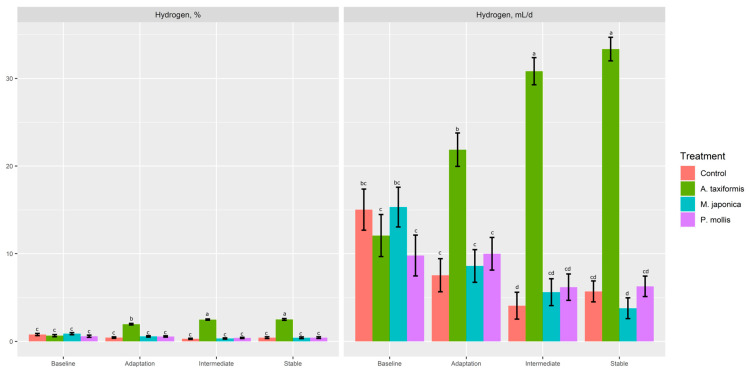
The effect of seaweed and phase on the percentage and production of H_2_ within a RUSITEC (*n* = 4). ^a–d^ Within variable, means without a common superscript differ (*p* ≤ 0.05).

**Table 1 animals-13-01643-t001:** Chemical composition of ingredients used in a rumen simulation technique (RUSITEC) to examine the effect of seaweeds on in vitro fermentation and methane production.

Ingredient	DM, %	OM, %DM	CP, %DM	NDF, %DM	EE, %DM	Bromoform, %DM
Barley silage	92.5	90.9	10.7	45.2	2.91	-
Barley straw	96.8	91.8	5.04	77.7	1.57	-
*Asparagopsis taxiformis*	96.2	52.6	18.4	50.3	0.37	0.517
*Mazzaella japonica*	19.9	71.3	20.7	66.9	0.43	ND
*Palmaria mollis*	12.4	61.2	21.5	35.6	0.34	ND

Abbreviations: DM = dry matter; OM = organic matter; CP = crude protein; NDF = neutral detergent fibre; EE = ether extract; ND = not detected.

**Table 2 animals-13-01643-t002:** Elemental analysis of the seaweeds used in a rumen simulation technique (RUSITEC) to examine the effect of seaweeds on in vitro fermentation and methane production.

Unit (mg/kg)	*Asparagopsis taxiformis*	*Mazzaella japonica*	*Palmaria mollis*
Macro-minerals			
Ca	37,397	4486	2626
K	14,397	26,432	108,198
Mg	7012	9815	4143
Na	56,641	56,936	28,389
P	1671	3022	4186
S	19,149	77,430	8782
Trace elements			
Al	7893.4	57.7	254.3
B	121.9	69.4	254.6
Co	2.5	<1.0	<1.0
Cr	22.8	7.8	4.5
Cu	4.8	1.0	2.3
Fe	5934	314	588
I	2580	15.7	58.1
Mn	93.6	14.2	13.6
Mo	2.1	1.0	0.5
Sb	<5.0	<5.0	<5.0
Se	<10.0	<10.0	<10.0
Zn	24.3	23.2	25.9
Toxic heavy metals			
As	16.1	9.7	9.7
Ba	9.2	0.3	0.9
Cd	<0.5	1.0	2.2
Hg	<10.0	<10.0	<10.0
Pb	<2.5	<2.5	<2.5
TI	<10.0	<10.0	<10.0

**Table 3 animals-13-01643-t003:** Effect of seaweed on in vitro nutrient disappearance, pH, volatile fatty acid, and ammonia production in a RUSITEC.

	Control	*Asparagopsis taxiformis*	*Mazzaella japonica*	*Palmaria mollis*	SEM	Treat	Phase	T × P
DMD, %	48.8 ^ab^	51.2 ^a^	48.8 ^ab^	46.9 ^b^	0.78	<0.01	<0.001	0.20
OMD, %	53.4	50.0	55.3	55.9	1.44	0.05	0.21	0.04
CPD, %	72.5	70.8	71.6	72.7	0.65	0.17	0.12	0.43
NDFD, %	34.0	28.1	36.8	36.7	2.04	0.03	0.06	0.05
Effluent, mL	648.1 ^ab^	633.4 ^b^	645.1 ^ab^	650.4 ^a^	4.23	0.03	0.03	0.08
pH	6.9	6.9	6.9	6.9	0.01	0.91	<0.001	0.36
VFA, mol/100 mol								
Acetate	53.8 ^a^	49.3 ^b^	53.5 ^a^	54.1 ^a^	0.60	<0.001	<0.001	0.26
Propionate	26.2	21.7	25.9	26.4	0.33	<0.001	<0.001	<0.001
Butyrate	13.0 ^b^	16.8 ^a^	13.2 ^b^	12.6 ^b^	0.56	<0.001	<0.001	0.12
BCVFA	1.22	1.63	1.21	1.27	0.021	<0.001	<0.001	<0.001
Caproate	0.64	1.75	0.69	0.65	0.080	<0.001	<0.001	<0.001
Valerate	5.15	7.72	5.24	4.84	0.422	<0.001	<0.001	<0.001
Total VFA, mmol/d	64.3	59.7	65.5	66.5	2.14	0.16	0.01	0.01
A:P	2.08	2.29	2.04	2.06	0.034	<0.001	<0.001	<0.001
NH_3_, mmol/d	3.31	3.16	3.28	3.40	0.087	0.30	<0.001	0.58

Abbreviations: DMD = dry matter disappearance; OMD = organic matter disappearance; CPD = crude protein disappearance; NDFD = neutral detergent fibre disappearance; VFA = volatile fatty acids; BCVFA = branched-chain VFA; A:P = acetate:propionate. ^a,b^ Values within a row with different superscripts differ significantly at *p* ≤ 0.05 for treatment, with no treatment x phase interaction.

**Table 4 animals-13-01643-t004:** Effect of seaweed on in vitro gas production in a RUSITEC.

	Control	*Asparagopsis taxiformis*	*Mazzaella japonica*	*Palmaria mollis*	SEM	Treat	Phase	T × P
Gas, mL/d	1450.2	1436.2	1523.8	1649.7	115.03	0.55	<0.001	0.61
CH_4_, %	3.75	1.45	3.60	4.08	0.238	<0.001	<0.001	<0.001
CH_4_, mL/d	59.0	20.2	51.8	65.9	4.59	<0.001	<0.001	<0.001
CH_4_, mg/d	41.6	13.9	36.7	45.9	3.06	<0.001	<0.001	<0.001
CH_4_, mg/g DMd	8.00	2.80	6.91	8.59	0.450	<0.001	<0.001	<0.001
CH_4_, mg/g DMi	4.15	1.39	3.67	4.58	0.306	<0.001	<0.001	<0.001
CO_2_, %	27.7 ^ab^	25.0 ^b^	26.2 ^ab^	29.8 ^a^	1.12	0.05	<0.001	0.24
CO_2_, mL/d	416.7 ^ab^	317.1 ^b^	386.3 ^ab^	474.8 ^a^	26.55	0.01	<0.001	0.50
H_2_, %	0.48	1.90	0.55	0.49	0.053	<0.001	0.06	<0.001
H_2_, mL/d	8.08	24.52	8.32	8.06	1.086	<0.001	0.72	<0.001

Abbreviations: DMd = dry matter digested; DMi = dry matter incubated. ^a,b^ Values within a row with different superscripts differ significantly at *p* ≤ 0.05 for treatment, with no treatment × phase interaction.

## Data Availability

The raw data supporting the conclusions of this article will be made available by the authors, without undue reservation.

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
