# Peer review of "Evaluation of Rumen Fermentation and Microbial Adaptation to Three Red Seaweeds Using the Rumen Simulation Technique"

_animals, 2023, doi:10.3390/ani13101643_

Round 1

Reviewer 1 Report

This is an interesting and rigorous experiment. I suggest minor corrections:

1.       Added error bars to all histograms in the manuscript

2.       In Statistical Analysis, add the statistical model formula

3.       50:50 Barley straw and barley silage as fermentation substrates. Can it satisfy the rumen microbial reproduction?

Author Response

Dear Reviewer,

Thank you for taking the time to review our manuscript. Please find below responses to your comments.

  1. Added error bars to all histograms in the manuscript

Response: Error bars have now been included in the figures and all figures have been updated.

  1. In Statistical Analysis, add the statistical model formula

Response: We would prefer to keep the written statistical model in the current form as it makes it easier to read than writing out the formula. However, we can change this if required by the reviewer.

  1. 50:50 Barley straw and barley silage as fermentation substrates. Can it satisfy the rumen microbial reproduction?

      Response: Yes, the barley silage is enough to provide microbial protein synthesis and production. This diet was chosen to represent a high forage diet, and is similar to the quality of diets that cattle graze over the winter in Canada.

Reviewer 2 Report

Dear authors,

the manuscript is engaging and interesting. There is quality and effort in this work, and since there is much interest in the use of macroalgae or seaweed in ruminant diets, it will give more insight into studying the effect of the three red seaweeds Asparagopsis taxiformis, Mazzaella japonica, and Palmaria mollis on in vitro fermentation and gas production, and microbial adaptation.

Major comments:

Results:

I have a few comments regarding the presented results in Table 3 and Table 4. If superscript letters define the significant differences based on the treat P value then:

Table 3: I believe superscripts are missing in the NDFD, Propionate, BCVFA, Caproate, Valerate, A:P, and NH3.

In addition, if superscript letters stand for the significant differences based on treat P value then as I seem to understand you found significant differences in most of the parameters. Therefore, please mention in the text (in the results section) the results of the significant differences of Table 4 measured parameters.

Table 4: If my abovementioned assumption regarding the superscript letters is correct, then please include superscript letters in some of the presented parameters that seem to differ significantly.

Figures:

Fig. 2: Why did you choose to present only adaptation and stable phases in OMD and NDFD? Also, please include the units in the presented parameters.

Fig 3b: Please include the unit of the A:P.

Discussion:

L313-316. I believe you need to support this statement also with reference(s).

L343-344. You can support this sentence with:  https://doi.org/10.3389/fmicb.2023.1104667

L387-390: If possible discuss a bit more with references that evaluated the impact of A. taxiformis in rumen microbiota. You can link it with: https://doi.org/10.3389/fmicb.2023.1104667.

Conclusion:

Regarding the final statement I think it would be wise to include a brief discussion (1-2 sentences in the discussion of the results of the VFA) regarding the possible impact of your results on performance.

Minor spelling:

L24: “an” adaptation phase, “an” intermediate phase

L45: has à have

L52: delete “over”

L159: You wrote consecutively 2 times the phrase “with a”

L332: Probably Ruminococcus not Ruminococcis?

L339: Add “S”: Similarly,

L347: from “the” immediate inclusion. Probably you need to add “the” there.

Author Response

Dear Reviewer,

Thank you for taking the time to review our manuscript. Please find responses to your comments below. We agree with all suggested changes.

Major comments:

Results:

I have a few comments regarding the presented results in Table 3 and Table 4. If superscript letters define the significant differences based on the treat P value then:

Table 3: I believe superscripts are missing in the NDFD, Propionate, BCVFA, Caproate, Valerate, A:P, and NH3.

In addition, if superscript letters stand for the significant differences based on treat P value then as I seem to understand you found significant differences in most of the parameters. Therefore, please mention in the text (in the results section) the results of the significant differences of Table 4 measured parameters.

Table 4: If my abovementioned assumption regarding the superscript letters is correct, then please include superscript letters in some of the presented parameters that seem to differ significantly.

Response: Due to the complexity of showing treatment and phase interaction responses within a table, we just used superscripts when there was a treatment phase alone (no a T x P effect). I have now defined this within the table endnotes. Thank you for bringing this to our attention, hopefully the superscript allocation and results now make more sense in this way.

Figures:

Fig. 2: Why did you choose to present only adaptation and stable phases in OMD and NDFD? Also, please include the units in the presented parameters.

Response: We have included this within the manuscript: Insufficient sample was available for baseline and intermediate phases, as samples were used for microbial profiling analysis, published elsewhere [9].(Lines 146-148). We have also added ‘%’ to Figure 2.

Fig 3b: Please include the unit of the A:P.

Response: A:P is a ratio so there is no unit required.

Discussion:

L313-316. I believe you need to support this statement also with reference(s).

Response: I have included a reference as suggested. L343-344.

You can support this sentence with:  https://doi.org/10.3389/fmicb.2023.1104667

Response: We have included this reference now. Thank you. L387-390.

If possible discuss a bit more with references that evaluated the impact of A. taxiformis in rumen microbiota. You can link it with: https://doi.org/10.3389/fmicb.2023.1104667.

Response: Have now included – (Lines 364-366) : This is further verified by a companion study that found that inclusion of A. taxiformis had a significant impact on the microbiome, including a large reduction in all major archaeal species [9].

Lines 376-378: . This conclusion is verified by O’Hara [9], who demonstrated that A. taxiformis inhibited major VFA producing bacteria including species of Fibrobacter and Ruminococcus

Conclusion:

Regarding the final statement I think it would be wise to include a brief discussion (1-2 sentences in the discussion of the results of the VFA) regarding the possible impact of your results on performance.

Response (412-416): These findings may indicate that feeding A. taxiformis to ruminants at a dose rate that results in a large decrease in CH4 production may alter rumen metabolism in a manner that restricts production performance through reduced VFA synthesis.

Minor spelling:

L24: “an” adaptation phase, “an” intermediate phase

L45: has à have

L52: delete “over”

L159: You wrote consecutively 2 times the phrase “with a”

L332: Probably Ruminococcus not Ruminococcis?

L339: Add “S”: Similarly,

L347: from “the” immediate inclusion. Probably you need to add “the” there.

Response: Thank you for picking up on these mistakes. I have corrected all listed.

Kind regards!